# Clinical and Genetic Characteristics of a Patient with Cystic Fibrosis with a Complex Allele [E217G;G509D] and Functional Evaluation of the CFTR Channel

**DOI:** 10.3390/genes14091705

**Published:** 2023-08-28

**Authors:** Elena Kondratyeva, Yuliya Melyanovskaya, Anna Efremova, Mariya Krasnova, Diana Mokrousova, Nataliya Bulatenko, Nika Petrova, Alexander Polyakov, Tagui Adyan, Valeriia Kovalskaia, Tatiana Bukharova, Andrey Marakhonov, Rena Zinchenko, Elena Zhekaite, Artem Buhonin, Dmitry Goldshtein

**Affiliations:** Research Centre for Medical Genetics, 115522 Moscow, Russia

**Keywords:** cystic fibrosis (CF), complex allele, targeted therapy, intestinal current measurements (ICM), intestinal organoids

## Abstract

The intricate nature of complex alleles presents challenges in the classification of *CFTR* gene mutations, encompassing potential disease-causing, neutral, or treatment-modulating effects. Notably, the complex allele [E217G;G509D] remains absent from international databases, with its pathogenicity yet to be established. Assessing the functionality of apical membrane ion channels in intestinal epithelium employed the intestinal current measurements (ICM) method, using rectal biopsy material. The effectivity of CFTR-targeted therapy was evaluated using a model of intestinal organoids of a patient harboring the genotype F508del/[E217G;G509D]. ICM analysis revealed diminished chloride channel function. Remarkably, [E217G;G509D] presence within intestinal organoids correlated with heightened residual CFTR function. Employing CFTR modulators facilitated the restoration of the functional CFTR protein. This multifaceted study intertwines genetic investigations, functional analyses, and therapeutic interventions, shedding light on the intricate interplay of complex alleles within CFTR mutations. The results highlight the potential of targeted CFTR modulators to restore functional integrity, offering promise for advancing precision treatments in cystic fibrosis management.

## 1. Introduction

Cystic fibrosis (CF) is an autosomal recessive disease that affects about 105,000 people in the world, making it one of the most common genetic diseases (https://www.cff.org/intro-cf/about-cystic-fibrosis, accessed on 24 August 2023). CF is associated with mutations in the cystic fibrosis transmembrane conduction regulator (*CFTR*) gene, which encodes a protein that regulates the transport of chlorine ions through cell membranes.

Currently, more than 2000 genetic variants of CFTR have been determined, some of them are rare, and their role in the development of CF and clinical outcomes has not been sufficiently studied [1]. Cystic fibrosis is characterized by a wide clinical polymorphism from a relatively mild course of the disease with mono-symptomatic manifestations to severe multi-organ lesions [2]. The prevalence of CF in European countries is on average 1 per 7000 newborns [3], in the Russian Federation, 1 per 8000–10,000 newborns, with variation in values by year and between populations and federal districts [2]. The spectrum and frequency of mutations in the CFTR gene vary widely in different populations and ethnic groups. As of 2023 on the website of the CFTR 2 international project (https://cftr2.org, accessed on 24 August 2023) 719 pathogenic variants of the nucleotide sequence of the CFTR gene are presented. The Russian national register of patients with CF remains a wide range of genetic variants. 233 variants with a frequency from 0.01% to 51.5% were recorded, 47 variants are not described in international genetic databases, and the clinical significance of some variants has not been established. In this regard, the identification and description of rare variants of the CFTR gene are crucial for understanding their pathogenetic significance. Identification of genetic variants makes it possible, taking into account the modern possibilities of pathogenetic therapy, to treat patients with CF. Among the identified 233 genetic variants, the largest number of variants of the CFTR gene in Russian patients (116) belong to class I disorders, 6 variants belong to class II, 5 variants belong to class III, 11 variants belong to class IV, 11 variants belong to class V, 1 variant belongs to class VI, the class is not established in 84 variants.

For the possibility of pathogenetic therapy, it is also important to determine the class of mutations and the type of genotype (severe, mild) in patients, depending on age.

The National Registry of CF patients in the Russian Federation has been maintained since 2011 and includes detailed genetic, diagnostic, clinical and therapeutic information about patients with CF. Since the publication of the first edition, the effectiveness of genetic examination has increased, leading to an increase in the total frequency of identified alleles from 80% to 89.9% in 2020 [4], due to the optimization of genetic research, including sequencing. Since 2014, information about complex alleles has become available in the register. The 2021 register indicates the allelic frequency of three complex alleles [5]. The selection of therapy for patients with cystic fibrosis also complicates the presence of complex alleles. Complex alleles result from the combination of two or more genetic variants in the CFTR gene in the cis position. Each of the variants in the complex allele itself can be pathogenic, have varying clinical significance or not lead to the development of cystic fibrosis. Complex alleles of the CFTR gene can affect the effectiveness of targeted therapy, while the mechanisms can be different: from increased pathogenicity with the simultaneous presence of two mutations on the same allele (for example, causing increased proteolysis of the CFTR protein) before changing the binding and, therefore, the effectiveness of the CFTR modulators. Most complex alleles are rare or unique [6].

In this article, we describe the clinical picture of cystic fibrosis in a patient with the newly identified complex allele p.[E217G;G509D] and present the results of a study of the CFTR function obtained using the intestinal current measurements method and the forskolin test on intestinal organoids. Intestinal organoids represent the key structural and functional features of the intestinal epithelium from which they were obtained from biopsy specimens. Organoids are multicellular self-organizing closed structures of single-layer epithelium, inside which there is a cavity called a “lumen”. On the membrane of intestinal organoids, facing inside the lumen, the CFTR channel is located. The forskolin-induced swelling (FIS) on intestinal organoids allows for personalized study of residual functional activity of the CFTR channel, as well as determination of the impact of targeted drugs on restoration of CFTR channel function. The results of the FIS may be a basis for prescribing targeted therapy to patients with cystic fibrosis. The main advantages of organoids in comparison with the model of cell lines are the possibility of a personalized approach to the therapy and diagnosis of a disease, as well as the long-term maintenance of a constant (transplanted) culture, the solution of the problem of species-specific differences, a full-fledged imitation of the cellular composition of an organ or tissue, from which organoids are obtained. 

The complex allele p.[E217G;G509D] includes the polymorphic variant E217G (c.650A>G, p.Glu217Gly) with allelic frequency 0.4583% in GnomAD_v.2.1.1 population database and the variant G509D (c.1526G>A, p.Gly509Asp) which was not registered in GnomAD. According to the literature, the missense variant E217G (replacement of glutamic acid with glycine) is considered as a genetic variant of uncertain value. The G509D variant is rare and leads to the replacement of glycine with aspartic acid in the nucleotide-binding domain of NBD1. At the time of the study, the complex allele [E217G;G509D] was not described in any of the international databases and the pathogenicity of this variant has not been established.

The aim is to determine the effect of the rare allele [E217G;G509D] on the pathogenesis of CF and clinical outcomes, with potential consequences for genetic counseling and personalized appointment of CFTR modulators. Some rare genetic variants of the *CFTR* gene may be associated with other diseases, such as bronchiectasis, pancreatic insufficiency and male infertility (CFTR-RDs).

## 2. Materials and Methods

The study was focused on the medical history of a male patient born in 2017 with the genotype [E217G;G509D]/F508del. The diagnostic has been completed according to the clinical guidelines [7].

### 2.1. Genetic Diagnostics

The Genomic DNA Purification Kit Wizard ^®^ (Promega, Madison, WI, USA) was used to isolate the patient’s DNA from whole blood. For the purpose of determining the genotype of the patient, the entire analysis of the CFTR gene (NM_000492.4) was performed by Sanger sequencing on an ABI 3500 instrument (Thermo Fisher Scientific, Waltham, MA, USA) in accordance with the manufacturer’s protocol. DNA diagnostics was performed according to the consensus algorithm “Cystic fibrosis: definition, diagnostic criteria, therapy”, section “Genetics of cystic fibrosis. Molecular genetic diagnostics in cystic fibrosis” [8]. To determine the coupling phases, exons 6 and 11 of the CFTR gene (NM_000492.4) were investigated by Sanger sequencing in parents.

### 2.2. Intestinal Current Measurements (ICM)

The study using the ICM method was conducted according to the European Standard operating procedures V2.7_26.10.11 (SOP) [9] according to the following algorithm. At the first stage, each recirculation chamber was calibrated separately on the VCC MC 8B421 (Physiological Instrument, San Diego, CA, USA). Physical factors were taken into account, such as the presence of air in the contact tips with agar and the resistance of the liquid, as well as environmental factors: the absence of vibrations near the equipment, accidental contacts with electrodes, and the absence of extraneous working devices in the office. At the second stage, after calibration of the device, rectal biopsy material was placed in the chamber. Biopsy samples were collected using Olympus Disposable EndoTherapy EndoJaw Biopsy forceps equipment (model #FB-23OU), Japan, according to the instructions. The size of the biopsy was 3–5 mm. The biopsy material was placed in a special slider, which was then inserted into the camera. The chambers were filled with Meyler buffer solution. The buffer was prepared before the study, it included: 105 mm NaCl, 4.7 mM KCl, 1.3 mM CaCl_2_·6H_2_O, 20.2 mM NaHCO_3_, 0.4 mM NaH_2_PO_4_·H_2_O, 0.3 mM Na_2_HPO_4_, 1.0 mM MgCl_2_·6H_2_O, 10 mM HEPES and 10 mM D-glucose, as well as 0.01 mM indomethacin. The registration of the study began with the recording of the basal short-circuit current (µA/cm^2^) (pre-amiloride stage). At the third stage, stimulants were added in the following sequence: amiloride (sodium channel), forskolin/IBMX (chloride channel), genistein (chloride channel), carbachol (calcium channel), DIDS (anion transport) and at the end—histamine (calcium channel). All reagents used are from Sigma-Aldrich (Merck), Germany. Following the recording of the basal short-circuit current, the study was concluded. The control group consisted of healthy volunteers. Patients with cystic fibrosis homozygous for F508del were included in the comparison group (F508del/F508del) [10]. The Scientific and Clinical Department of Cystic Fibrosis of the Research Centre for Medical Genetics was where the study was conducted (Head—Prof. Kondratyeva E.I.).

### 2.3. Human Intestinal Organoids Culture

Cultures of intestinal organoids were obtained in accordance with protocols developed by J.M. Beekman [11,12,13,14]. The method of obtaining intestinal organoids from rectal biopsies at the RCMG was described in detail earlier in [6]. The sampling of the patient’s biological material was carried out after signing an informed voluntary consent. Individual crypts were isolated from rectal biopsies, for this purpose incubation was carried out with a solution of 10 mM EDTA (Thermo Fisher Scientific, USA). Crypts were embedded in “Matrigel” (Corning, NY, USA) and seeded into 24-well plates (Thermo Fisher Scientific, USA). After the polymerization of the “Matrigel”, a culture medium was added. The composition of the medium is indicated in [13]. Organoids were split once every 7 days by mechanical destruction of large budding structures into small fragments.

### 2.4. Forskolin-Induced Swelling (FIS) Assay

For the FIS assay, organoids were seeded into 96-well plates. After 24 h, the organoids were stained with Calcein AM (Biotium) and stimulated with forskolin (Sigma-Aldrich, USA) at concentrations of 0.128 and 5 µM. The processing lasted for 60 min, and the “fixed” fields were photographed using the Observer. D1 fluorescence microscope (Zeiss, Germany) at 10-min intervals. CFTR correctors VX-809, VX-661 and VX-445 were all added at a concentration of 3.5 µM (Selleckchem, Houston, TX, USA) at the stage of organoids seeding, and CFTR potentiator VX-770 (3.5 µM; Selleckchem, USA)—simultaneously with forskolin. Quantitative analysis of the FIS was carried out using the Image J program (v1.52n state version). The Sigma Plot 12.5 program was used for plotting.

## 3. Results

### 3.1. Description of the Clinical Picture

A child born in 2017, was a boy with a diagnosis of cystic fibrosis, mainly pulmonary form (E84.0.). Chronic bronchitis. Genetic diagnosis: [E217G;G509D]/F508del. Microbiological analysis: intermittent *Pseudomonas aeruginosa*, *Stenotrophomonas maltophilia* (2019).

The diagnosis of cystic fibrosis was verified by positive CF neonatal screening (IRT1—55.6 (norm < 70 ng/mL on day 5), IRT2–43.8 ng/mL (norm < 40 ng/mL); positive result of sweat test, conductivity on a Nanodact apparatus 83-81-63-65 mmol/L (norm < 50 mmol/L). The results are borderline and positive; characteristic clinical picture (cough, salty sweat); genetic examination: [G509D;E217G]/F508del; coprology: no neutral fat; pancreatic elastase over 500 mcg/g of feces; results of microbiological examination—*P. aeruginosa*.

Family history: The father had acute pancreatitis in May 2020, which required hospitalization. The DNA diagnosis of the *CFTR* gene in the mother and father was carried out. The father has a complex allele—[E217G;G509D] in a heterozygous state, the mother has a genetic variant of F508del in a heterozygous state.

Anamnesis: from 3 normal pregnancies, 2 urgent deliveries for 38 weeks, weight is 3.5 kg and the height is 52 cm, meconium departed on time, physiological jaundice lingering (3 weeks). Stool 5–6 times a day while breastfeeding, without visible steatorrhea, respiratory syndrome—absent.

From 1 month before the introduction of complementary foods—constipation. Breastfeeding up to 4 months, from 5 months on milk formula. For 1 year of life, acute respiratory infections—3 times, antibacterial therapy once. In the first year of his life, he gained 8 kg weight, 25 cm tall. The stool in the dynamics is regular, there are no constipation. He grows and develops according to age.

During the examination in the first year of life:

Abdominal ultrasonography—hepatosplenomegaly, diffuse liver changes; lung radiography—without pathology. Pancreatic elastase > 500 mcg/g of feces. Coprology—there is no neutral fat. Blood Biochemistry: AST—23 U/L (N 15.0–60.0 U/L), ALT 53—U/L (N 10–40 U/L), ALP—236 U/L (N 156.0–369.0 U/L).

From birth, he received basic therapy: Dornase α inhalation, kinesitherapy, vitamin therapy: vit. E, D3, A, ursodeoxycholic acid.

Primary colonization of the respiratory tract with *P*. *aeruginosa* occurs at the age of 1 year and 6 months. Therapy of primary colonization of *Pseudomonas aeruginosa*: inhaled tobramycin, 3 courses. In 2019, again: *P. aeruginosa* 10^5^ (titer growth in dynamics). On this occasion, tobramycin therapy was carried out for a year. In the future, there will be no *P*. *aeruginosa* detected.

In April 2019, the primary colonization of *S. maltophilia* 10^5^, sensitivity only to fluoroquinolones, and resistance to other drugs. The child received ciprofloxacin, there is no further growth. Eradication was carried out.

Lung radiography in 2018 and 2019 without pathology.

During the examination in 2019–2020: fecal elastase 09/2019—more than 500 mcg per g of feces, from 12/2019–375 mcg/g of feces; hematologic, urinary test, blood biochemistry—without pathology; microbiological examination—there is no pathogenic flora.

During Sanger sequencing of the entire CFTR gene, the child was identified with the genotype F508del/[E217G;G509D],which includes the complex allele [E217G;G509D], which consists of the genetic variant G509D (c.1526G>A, p.Gly509Asp) and polymorphic variant E217G (c.650A>G, p.Glu217Gly), which according to the literature is considered as benign, not leading to cystic fibrosis (Figure 1).

Considering the presence of a previously undescribed genetic variant of G509D, a weakly positive result of IRT2—43.8 ng/mL (N < 40), positive or borderline results of a sweat test on a Nanodact (83-81-63-65-63 mmol/L), preserved pancreatic function according to pancreatic elastase data of >500 mcg/g of feces, the absence of intestinal and respiratory syndromes up to 4 years old, while the child has an early infection with *S. maltophilia* and *P. aeruginosa*, it was decided to conduct a ICM method and a forskolin test on intestinal organoids to clarify violations of the CFTR channel and selection of targeted therapy.

### 3.2. Evaluation of the Functional Activity of the CFTR Channel by the ICM Method

The conducted study by the ICM method showed the following results: the short-circuit current density (ΔI_SC_) in response to the introduction of amiloride (stimulation of sodium channels) was −20.33 ± 4.95 µA/cm^2^. The change in ΔI_SC_ in response to the introduction of forskolin (stimulation of chloride channels) was 25 ± 3.37 µA/cm^2^, which corresponded to a reduced function of the chlorine channel. In response to the introduction of carbachol, the ΔI_SC_ changes in the negative direction and was 22 ± 1.97 µA/cm^2^. In response to the introduction of histamine, the ΔI_SC_ changes in the negative direction, which reflects the entry of potassium ions into the cells. At the same time, the current density was 15.67 ± 2.41 µA/cm^2^ (Table 1 and Figure 2).

When analyzing the data, it was found that the response to amiloride in a patient with the G509D variant was lower than in the F508del/F508del group but higher than in the control group. When analyzing the response to forskolin: the patient’s response was equal to the control group and higher than in the F508del/F508del group. However, when histamine was added, the response in the patient with the G509D variant was lower than in the other groups (F508del/F508del and the control group) and the response was in the negative direction, which reflects the entry of potassium ions into the cells as in cystic fibrosis.

The test indicates a reduced function of the CFTR channel and confirms the pathogenic significance of this mutation and the diagnosis of cystic fibrosis.

### 3.3. Evaluation of the Effect of CFTR Modulators on the Restoration of the CFTR Channel Function on the Model of Intestinal Organoids

From rectal biopsies of a patient with genotype [E217G;G509D]/F508del, a constant culture of intestinal organoids was obtained to assess the effect of targeted drugs on the function of the CFTR channel. To begin with, the morphology of the patient’s organoids was compared with control cultures from a patient with the F508del/F508del genotype and a healthy volunteer (Figure 3). Organoids with a working CFTR channel have a spherical shape, thin walls and large lumens relative to the total size of the organoid (Figure 3, wt/wt organoids). [E217G;G509D]/F508del organoids, like the F508del/F508del control, were characterized by an irregular shape and a smaller size of the internal cavity compared to samples from healthy donors. Morphological signs of organoids of a patient with a complex allele [E217G;G509D] indicate a violation of the functional activity of the CFTR protein (Figure 3).

In a culture of intestinal organoids, it was shown that the presence of [E217G;G509D] is accompanied by a residual CFTR function, since when exposed to forskolin, organoids respond with swelling (Figure 4 and Figure 5), for example, with 1 h exposure to 5 µM of forskolin, the size of organoids increases by 50% (Figure 4B). It is due to the presence in the genotype of the complex allele [E217G;G509D] since the second allele with the F508del variant causes an almost complete loss of the functional CFTR protein (for comparison—F508del/F508del control, Figure 4C).

The VX-770 potentiator promotes the restoration of the functional CFTR protein—with the combined effect of VX-770 and 5 µM of forskolin, organoids increase more than twice (by 130%) after 1-h incubation (Figure 4B and Figure 5). Since the genotype [E217G;G509D]/F508del is characterized by high preservation of the CFTR function, with strong forskolin stimulation (5 µM) against the background of a strong response to forskolin, there is no additive effect from the combined use of a potentiator and correctors—under all exposure conditions (VX-770, VX-770 + VX-809, VX-770 + VX-661, VX-770 + VX-661 + VX-445) the answer is 115–130% (Figure 4B and Figure 5).

The effect of CFTR modulators on the restoration of functional CFTR in the presence of high residual function is more pronounced with weak stimulation by forskolin. Thus, at 1 h exposure to 0.128 µM forskolin, a significant increase in organoid swelling is observed with the combined effect of the potentiator VX-770 and all correctors compared with the effect of only the potentiator or corrector (exemplified by VX-809) (Figure 4A).

## 4. Discussion

The complex allele [E217G;G509D] was detected for the first time in a patient with CF. The pathogenicity of the genetic variant E217G is poorly understood. The diagnosis of cystic fibrosis was established on the basis of a positive result of neonatal screening for IRT2, and positive and borderline values of the sweat test. The course of cystic fibrosis was characterized by preserved pancreatic function, absence of respiratory manifestations for up to 6 years, and intermittent Pseudomonas aeruginosa infection in 2 years. The ICM method showed a reduced function of the chloride channel (ΔI_SC_ for forskolin 20.5 ± 3.37 µA/cm^2^, in the control group −25.78 ± 4.41 µA/cm^2^, in homozygotes −3.06 ± 0.89 µA/cm^2^).

For patients with the complex allele [E217G;G509D] a therapeutic effect can be expected from all combined CFTR modulators, because VX-770, VX-809, VX-770 + VX-809, VX-770 + VX-661, VX-770 + VX-661 + VX-445 promotes the restoration of the functional CFTR protein.

It should be noted that the father of the child—the carrier of the complex allele [E217G; E509D]—had a history of acute pancreatitis at the age of 40, which required hospitalization and conservative therapy. It is known that carriers of a number of genetic variants of CFTR develop acute pancreatitis [18,19]. The father is recommended to continue the molecular genetic diagnosis of hereditary pancreatitis. Currently, the child continues to be monitored at the cystic fibrosis center.

The clinical picture of the patient with the genotype [E217G;G509D]/F508del, the results of functional tests (sweat samples and the ICM method) and the responses of intestinal organoids to forskolin and CFTR modulators correspond to the manifestations of “soft” mutations. Based on the results obtained, it can be assumed that this complex allele [E217G;G509D] belongs to mutations of class IV–VI.

For the first time, complex alleles of the *CFTR* gene and their phenotypic manifestations were described in 1991 [20]. Cases are described when each of the variants in the complex allele does not lead to the occurrence of CF, but their combined pathogenic effect is summed up and leads to the development of clinical manifestations of the disease [21]. It is thus described that one variant of Asp1270Asn (in combination with another pathogenic parent allele in the compound-heterozygote state) is not enough to cause CFTR channel dysfunction. While the complex allele p.[Arg74Trp;Asp1270Asn] had a slight decrease in the function of the chlorine channel, and p.[Arg74Trp;Val201Met;Asp1270Asn] caused CF.

In our clinical case, the pathogenicity of the genetic variant E217G is poorly understood. CFTR-France databases [22] and ClinVar [23] consider it not pathogenic. However, in combination with G509D (missense mutation), it causes CF, as this clinical case shows. And with a heterozygous carrier, as in the father, a CF-associated disease occurs—pancreatitis.

There are also opposite effects when the course of the disease becomes milder. Complex alleles in the *CFTR* gene contain at least two variants of the sequence of the same gene in the cis position, and each pathogenic variant can affect individual stages of CFTR biogenesis. The effect of complex alleles on the phenotype depends on the in-trans variant.

Complex alleles can influence the effectiveness of treatment with CFTR modulators, changing the effectiveness of the modulator, which must be taken into account when selecting targeted therapy [24,25]. It is described that the variant p.Leu467Phe does not cause CF, but reduces the amount of functional protein CFTR by 2 times compared to the norm, but when p.Leu467Phe is combined with p.Phe508del, there is no response to treatment with two-component correctors [15,26,27]. In this case, we observed the response to all modulators when using intestinal organoids. It is possible that complex alleles can increase the effectiveness of targeted therapy in a number of patients.

## 5. Conclusions

The results of the study make an additional contribution to understanding the features of the clinical course of cystic fibrosis in carriers of complex alleles. The complex allele [E217G;G509D] is characterized by a high residual function of the CFTR channel. For patients with the complex allele [G509D;E217G], a therapeutic effect can be expected from the use of all combined CFTR modulators.

## Figures and Tables

**Figure 1 genes-14-01705-f001:**
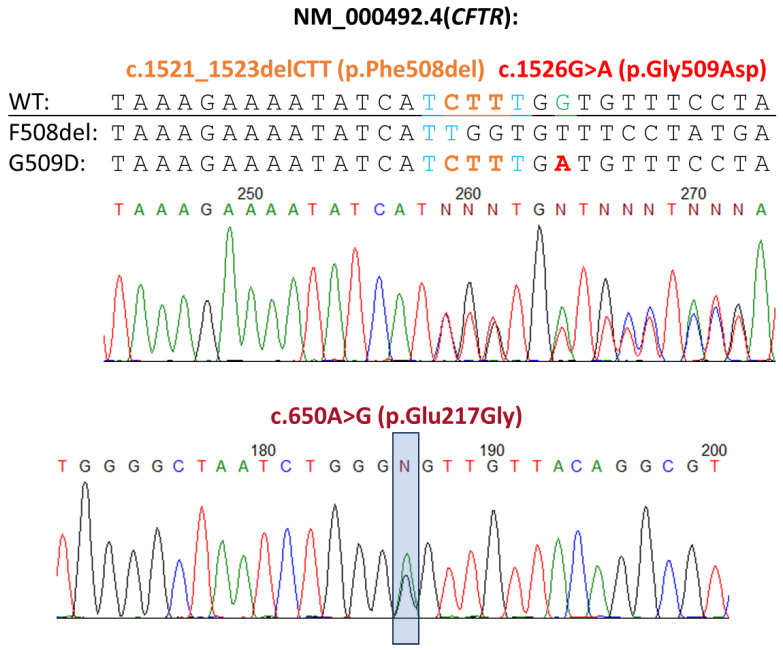
Chromatogram results of Sanger sequencing of the CFTR gene, demonstrating the presence of c.1521_1523delCTT (F508del) (highlighted in orange in top panel) in trans with c.1526G>A (G509D, p.Gly509Asp) (highlighted in red in top panel) and c.650A>G (E217G, p.Glu217Gly) (highlighted in dark red and squared in bottom panel) variants in the patient.

**Figure 2 genes-14-01705-f002:**
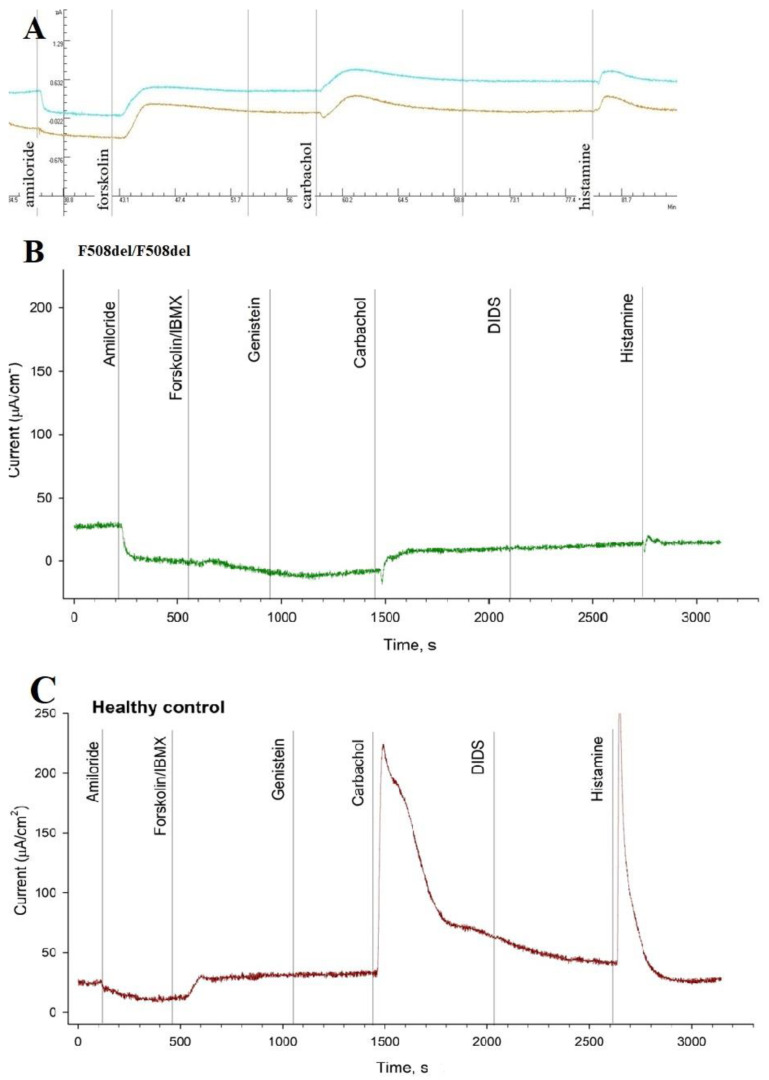
The ICM method. (**A**) Patient with genotype F508del/[E217G;G509D]. Note: With the introduction of amiloride, a response was obtained, with the addition of forskolin/IBMX, the response was reduced, and with the addition of histamine and carbachol, a change in the short-circuit current was observed in the negative direction. (**B**) Patient with genotype F508del/F508del. When amiloride was administered, there was a decrease in the short-circuit current (ΔI_SC_), but no changes were observed in response to the introduction of forskolin/IBMX, and a negative change in the short-circuit current was observed in response to the addition of histamine; (**C**) The control group. The addition of amiloride caused a decrease in ΔI_SC_, there was a significant increase in ΔI_SC_ in response to forskolin/IBMX, while the addition of histamine led to a change of short circuit current in the positive direction.

**Figure 3 genes-14-01705-f003:**
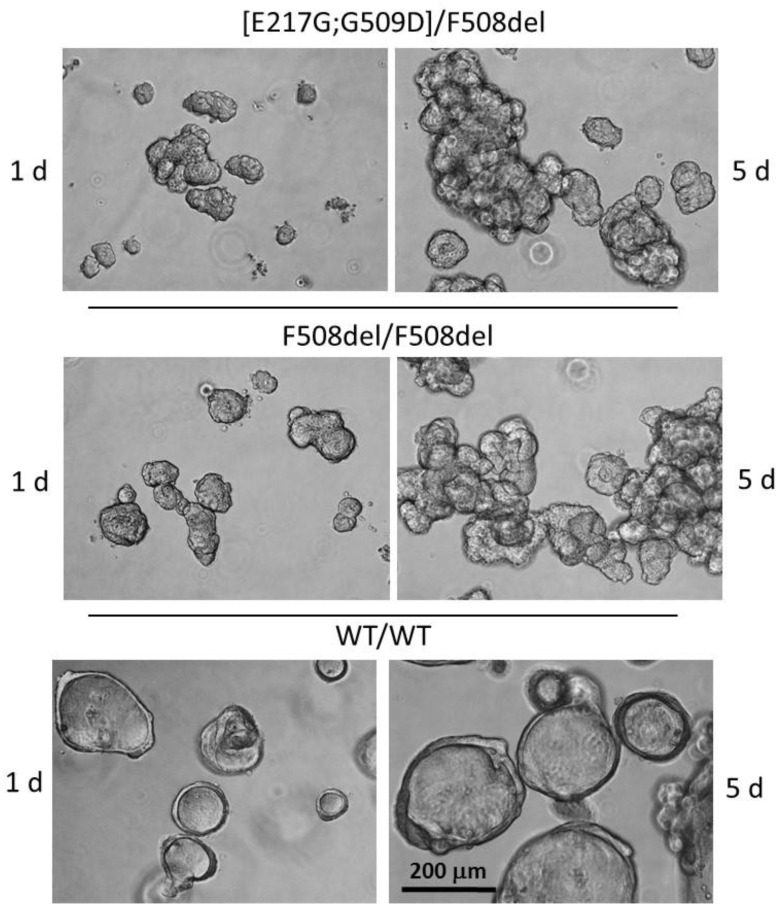
Morphology of intestinal organoid cultures from a patient with a complex allele [E217G;G509D] in comparison with control organoid cultures obtained from a healthy volunteer (wt/wt) and a patient with the genotype F508del/F508del.

**Figure 4 genes-14-01705-f004:**
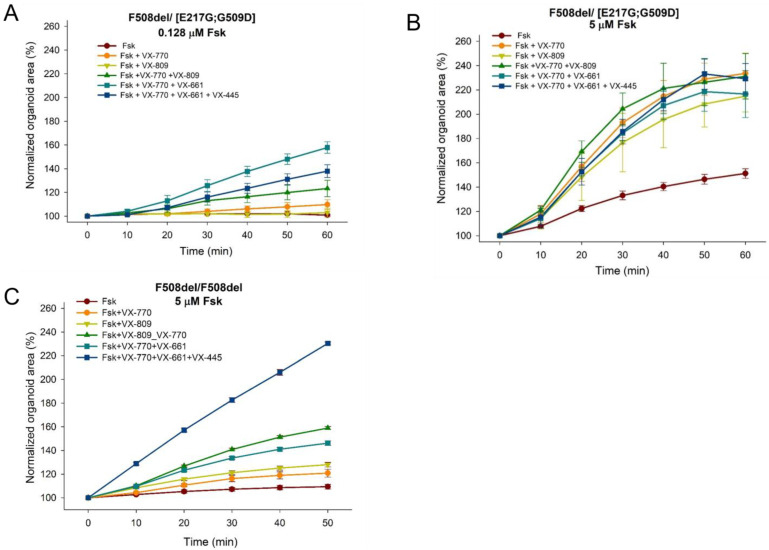
Results of quantitative evaluation of the effect of CFTR modulators on forskolin-induced swelling of intestinal organoids obtained from a patient with a complex allele [E217G;G509D] (**A**,**B**) and a patient with the genotype F508del/F508del (control) (**C**). The response of [E217G;G509D]/F508del organoids to CFTR modulators when exposed to 0.128 µM (**A**) or 5 µM (**B**) forskolin. The response of the control culture of F508del/F508del organoids to CFTR modulators when exposed to 5 µM forskolin (**C**).

**Figure 5 genes-14-01705-f005:**
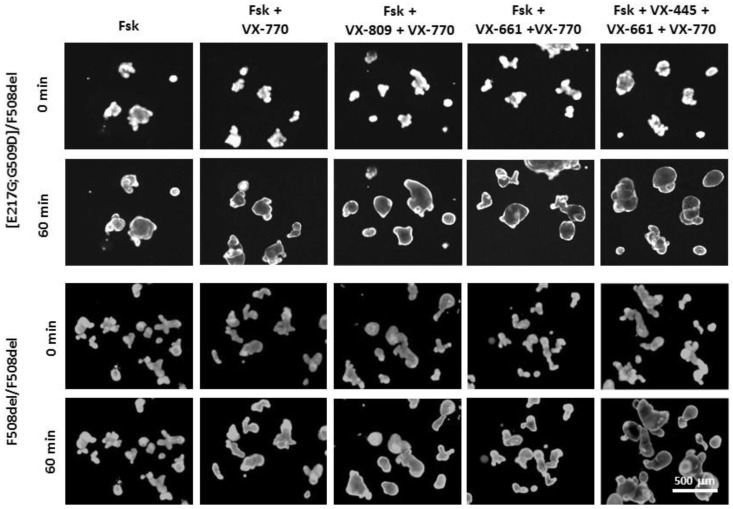
Representative images of intestinal organoids with [E217G;G509D] complex allele and F508del/F508del genotype (control) before and after treatment with 5 µM forskolin and CFTR modulators.

**Table 1 genes-14-01705-t001:** Indicators of short-circuit current density (ΔI_SC_) during the administration of stimulants in a patient carrying the complex allele [E217G;G509D] in the genotype.

ΔI_SC_, µA/cm^2^	Amiloride	Forskolin	Genistein	Carbachol	DIDS	Histamine
M ± m patient	−13.67 ± 4.56	20.5 ± 3.37	0	22 ± 1.97	0	15.67 ± 2.41
F508del/F508del [*, **]	−18.39 ± 5.62	3.06 ± 0.89	1.83 ± 0.35	-	1.83 ± 0.35	21.5 ± 5.46
Control group [*, **]	−8.98 ± 3.42	25.78 ± 4.41	2 ± 0.29	117.44 ± 4.32	1.8 ± 0.26	101.68 ± 10.99

Note: *—Clinical guidelines for Cystic Fibrosis 2021 [7]. **—[15,16,17].

## Data Availability

The corresponding author is able to provide the datasets used and/or analysed during the current study upon reasonable request.

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
