# Peer review of "Clinical and Genetic Characteristics of a Patient with Cystic Fibrosis with a Complex Allele [E217G;G509D] and Functional Evaluation of the CFTR Channel"

_genes, 2023, doi:10.3390/genes14091705_

Round 1

Reviewer 1 Report

Kondratyeva et al present here a case report of a patient with CF symptoms, F508del and the complex allele [E217G;G509D], which they describe for the first time. The importance of complex alleles and possible implications for, for example, responses to CFTR modulators has become increasingly clear. The authors describe clinical symptoms of this case, as well as ICM for diagnosis. They then proceed with growing of intestinal organoids, to evaluate the phenotype of these organoids and responses to CFTR modulators.

Major comments

Figure 2/3/4: perhaps these figures can be combined into one bigger figure. Also, the legends lack explanation about the experimental set-up. Can the authors explain the lack of response to forskolin in the healthy control tracing? Why was Carbachol not added in the F508del control?

Figure 6: The authors claim that the complex allele is responsive to all VX modulators tested, but it is impossible to make this claim based alone on the data presented here. Since, for example, elexacaftor (VX-445) was not tested without the other modulators.

Can the authors also specify why they chose not to work with the 0.8µM of FSK which has been shown by several groups to be the FSK concentration that best correlates with in patient responses?

I would also recommend to the authors to also discuss the importance of complex alleles in a bigger context and the implications of the current study for this. Besides, the current study did not allow to identify the effects of the individual parts of the complex allele. It would be good if they could expand on this a little.

Minor comments:

Line 26: please update the number of people with CF

Line 28 and others: please change gene names in italics

Line 45: it would be useful if the authors would include some numbers on the prevalence of these individual variants.

Line 48: could the authors specify the current classification of the G509D variant? Is it disease-causing?

Figure 1: could the authors annotate the top half of the figure as well (E217G)? Now it is not immediately clear what we are supposed to see here.

Line 183 and others: Chloride instead of chlorine

Figure 5: It is unclear to me why 1d timepoint is only shown for the [E217G;G509D]/F508del and not the others

Figure 7: I would suggest to add an image of F508del as well here.

I recommend the authors consult a native English speaker to adapt the language in this manuscript.

Author Response

Dear reviewer, thank you for your attention and thorough analysis of our work.

Major comments

  1. Figure 2/3/4: perhaps these figures can be combined into one bigger figure. Also, the legends lack explanation about the experimental set-up. Can the authors explain the lack of response to forskolin in the healthy control tracing? Why was Carbachol not added in the F508del control? Response 1: Figures 2-4 have been merged into one (#2). A note about the control group and F508del/F508del has been added to the legend. The response to forskolin was present in the control group, which is confirmed by the data presented in the table. Corrected the drawing, the answer to carbachol is shown in the figure.
  2. Figure 6: The authors claim that the complex allele is responsive to all VX modulators tested, but it is impossible to make this claim based alone on the data presented here. Since, for example, elexacaftor (VX-445) was not tested without the other modulators. Response 2: Thank you for your comment. Indeed, we show the results only for the VX-809 corrector (VX-661 was tested separately, VX-445 – not tested), since VX-809 и VX-661 correctors alone without a potentiator showed similar efficiency (data not shown). We decided not to provide data for each corrector separately, because they are not used as standalone drugs and as an example we only show results for corrector VX-809 The text was changed: « Thus, at 1 h exposure to 0.128 uM forskolin, a significant increase in organoid swelling is observed with the combined effect of the potentiator VX-770 and all correctors compared with the effect of only the potentiator or corrector»  to  «Thus, at 1 h exposure to 0.128 uM forskolin, a significant increase in organoids swelling is observed with the combined effect of the potentiator VX-770 and all correctors compared with the effect of only the potentiator or corrector (exemplified by VX-809)»
  3. Can the authors also specify why they chose not to work with the 0.8µM of FSK which has been shown by several groups to be the FSK concentration that best correlates with in patient responses? Response 3: Our team has a significant experience in performing FIS assay using 0,128 µM and 5 µM forskolin (https://doi.org/10.1016/j.gene.2020.145023; 3390/genes12060837; doi.org/10.3390/ijms24076351). According to our results “mild” CFTR variants, marked differences in CFTR-modulator effects are observed when using 0,128 µM forskolin (https://doi.org/10.3390/genes12060837 Figure 4; doi.org/10.3390/ijms24076351 Figure 5 genotype E92K/L138ins). In this study we used maximum  forskolin concentration of 5 µM, since for  “severe” genotypes the differences in CFTR-modulator effects are most prominent at this concentration (https://doi.org/10.1016/j.jcf.2018.07.001, https://doi.org/10.1016/j.gene.2020.145023 Figure 5B).
  4. I would also recommend to the authors to also discuss the importance of complex alleles in a bigger context and the implications of the current study for this. Besides, the current study did not allow to identify the effects of the individual parts of the complex allele. It would be good if they could expand on this a little. Response 4: done

Minor comments:

  • Line 26: please update the number of people with CF Response 1: According to the American Society for Cystic Fibrosis, there are approximately 105,000 CF patients worldwide. https://www.cff.org/intro-cf/about-cystic-fibrosis
  • Line 28 and others: please change gene names in italics Response 2: done
  • Line 45: it would be useful if the authors would include some numbers on the prevalence of these individual variants. Response 3: done

frequency unknown

  • Line 48: could the authors specify the current classification of the G509D variant? Is it disease-causing? Response 4: There is no information about the G509D variant. Based on our data, this variant can be considered pathogenic.
  • Figure 1: could the authors annotate the top half of the figure as well (E217G)? Now it is not immediately clear what we are supposed to see here. Response 5: The figure has been corrected
  • Line 183 and others: Chloride instead of chlorine Response 6: done
  • Figure 5: It is unclear to me why 1d timepoint is only shown for the [E217G;G509D]/F508del and not the others. Response 7: Thank you for noticing, we have changed the figure and provided images for wt/wt и F508del/F508del after 1 day of culture.
  • Figure 7: I would suggest to add an image of F508del as well here. Response 8: Thank you, we have replaced the figure and added images with F508del organoid responses to modulators.

Reviewer 2 Report

The present study has scientific merit and contributes to the advancement of studies in medical genetics. However, some points need to be improved:

1. Authors need to include more references in the introduction, as some statements are not referenced.

2. The authors need to be more succinct when describing the methodology, for example, the sentence: "Biopsy samples were collected using Olympus Disposable EndoTherapy EndoJaw Biopsy forceps equipment (model #FB- 23OU), according to the instructions" is written twice in the text.

3. Include in the methodology the suppliers/brands of reagents used in sample preparation.

4. Has the present study been approved by the local Ethics Committee?

5. The name of the microorganism must be written in italics, please make corrections throughout the text.

6. In the results section, the authors perform several comparisons with control groups. In this case, shouldn't they use statistical tests?

7. The discussion needs to be expanded for a better understanding of the importance of the study for the academic community.

Author Response

Dear reviewer, thank you for your attention and thorough analysis of our work

  1. Authors need to include more references in the introduction, as some statements are not referenced. Response 1: Done
  2. The authors need to be more succinct when describing the methodology, for example, the sentence: "Biopsy samples were collected using Olympus Disposable EndoTherapy EndoJaw Biopsy forceps equipment (model #FB- 23OU), according to the instructions" is written twice in the text. Response 2: Done
  3. Include in the methodology the suppliers/brands of reagents used in sample preparation. Response 3: Thank you. We have added data on reagent grades.
  4. Has the present study been approved by the local Ethics Committee? Response 4: The study and the form of informed voluntary consent were approved by the Ethics Committee of the "RCMG" of the Ministry of Education and Science of the Russian Federa-tion on October 15, 2018 (the chairman of the Ethics Committee is Prof. L. F. Kurilo).
  5. The name of the microorganism must be written in italics, please make corrections throughout the text. Response 5: done
  6. In the results section, the authors perform several comparisons with control groups. In this case, shouldn't they use statistical tests? Response 6: The comparison was carried out without statistical processing methods
  7. The discussion needs to be expanded for a better understanding of the importance of the study for the academic community. Response 7: We have expanded the discussion section
